# Normo- or Hypo-Fractionated Photon or Proton Radiotherapy in the Management of Locally Advanced Unresectable Pancreatic Cancer: A Systematic Review

**DOI:** 10.3390/cancers15153771

**Published:** 2023-07-25

**Authors:** Sally A. Elkhamisy, Chiara Valentini, Annika Lattermann, Ganesh Radhakrishna, Luise A. Künzel, Steffen Löck, Esther G. C. Troost

**Affiliations:** 1OncoRay-National Center for Radiation Research in Oncology, Faculty of Medicine and University Hospital Carl Gustav Carus, Technische Universität Dresden, Helmholtz-Zentrum Dresden-Rossendorf, 01307 Dresden, Germany; Sally.Abdelhaleem@ukdd.de (S.A.E.); annika.lattermann@ukdd.de (A.L.);; 2Department of Clinical Oncology and Nuclear Medicine, Faculty of Medicine, Mansoura University, Mansoura 35516, Egypt; 3Department of Radiotherapy and Radiation Oncology, Faculty of Medicine and University Hospital Carl Gustav Carus, Technische Universität Dresden, 01307 Dresden, Germany; 4The Christie Hospital NHS Foundation Trust, Manchester M20 4BX, UK; g.radhakrishna@nhs.net; 5National Center for Tumor Diseases (NCT), Partner Site Dresden, Germany: German Cancer Research Center (DKFZ), Heidelberg, Germany; Faculty of Medicine and University Hospital Carl Gustav Carus, Technische Universität Dresden, Dresden, Germany, and Helmholtz Association/Helmholtz-Zentrum Dresden—Rossendorf (HZDR), 01307 Dresden, Germany; 6German Cancer Consortium (DKTK), Partner Site Dresden, German Cancer Research Center (DKFZ), 69120 Heidelberg, Germany; 7Helmholtz-Zentrum Dresden-Rossendorf, Institute of Radiooncology-OncoRay, 01328 Dresden, Germany

**Keywords:** locally advanced pancreatic adenocarcinoma, photon therapy, SBRT, proton beam therapy, systematic review

## Abstract

**Simple Summary:**

Locally advanced pancreatic carcinoma (LAPC) is one of the most challenging tumors that requires multimodality management including chemotherapy and/or radiotherapy. Local radiotherapy toxicity is a fundamental problem for dose escalation, so advances in radiation therapy using different techniques or fractionations are being developed. We focus this systematic review on the outcome of the photon and proton radio (chemo) therapy with normo- or hypo-fractionation schedules in management of patients with locally advanced pancreatic carcinoma.

**Abstract:**

LAPC is associated with a poor prognosis and requires a multimodal treatment approach. However, the role of radiation therapy in LAPC treatment remains controversial. This systematic review aimed to explore the role of proton and photon therapy, with varying radiation techniques and fractionation, in treatment outcomes and their respective toxicity profiles. Methods: Clinical studies published from 2012 to 2022 were systematically reviewed using PubMed, MEDLINE (via PubMed) and Cochrane databases. Different radiotherapy-related data were extracted and analyzed. Results: A total of 31 studies matched the inclusion criteria. Acute toxicity was less remarkable in stereotactic body radiotherapy (SBRT) compared to conventionally fractionated radiotherapy (CFRT), while in proton beam therapy (PBT) grade 3 or higher acute toxicity was observed more commonly with doses of 67.5 Gy (RBE) or higher. Late toxicity was not reported in most studies; therefore, comparison between groups was not possible. The range of median overall survival (OS) for the CFRT and SBRT groups was 9.3–22.9 months and 8.5–20 months, respectively. For the PBT group, the range of median OS was 18.4–22.3 months. Conclusion: CFRT and SBRT showed comparable survival outcomes with a more favorable acute toxicity profile for SBRT. PBT is a promising new treatment modality; however, additional clinical studies are needed to support its efficacy and safety.

## 1. Introduction

Pancreatic adenocarcinoma is ranked seventh among cancer-related deaths throughout the world, since even with continuous advances in diagnosis and treatment, the 5-year OS remains only 9% [1]. At the first presentation, approximately 30% of patients are diagnosed with LAPC, of whom 50% are found to have metastatic disease. Consequently, only 20% of the patients are considered to have a resectable tumor [2]. Typical symptoms of patients with LAPC are abdominal pain, repetitive obstructive jaundice, and weight loss, which leads to more deterioration of patient general condition and quality of life [3]. 

According to the National Comprehensive Cancer Network (NCCN) guidelines, in most cases, management of LAPC starts with induction chemotherapy (IC), e.g., modified FOLFIRINOX or albumin-bound paclitaxel and gemcitabine [4]. Apart from systemic treatment, radiation therapy (RT) as opposed to surgery, is controversial due to treatment-related toxicity without improvement of local tumor control and overall survival [5]. However, intensified treatment with induction chemotherapy (IC) followed by intensified high-precision high-dose radiation therapy has been adopted in many trials as a new paradigm to increase the local control, downstage the primary tumor and make surgical resection feasible [6].

The anatomical location of the pancreas amidst the radiation-sensitive duodenum, stomach as well as large bowel, poses a challenge for radiation treatment. These neighboring structures prohibit dose escalation due to acute and long-term toxicity possibly introduced by radiotherapy [7]. Moreover, gastric filling and the respiratory cycle induce movement of the pancreas, in particular in superior–inferior direction, which should be estimated to insure high-precision radiation delivery [8]. Adaptive radiation delivery strategies, such as breath holding technique, abdominal compression and respiratory gating, attempt to decrease the internal target and planning target volumes (ITV and PTV, respectively) and thus spare organs at risk [9]. 

Intensity-modulated radiation therapy (IMRT) using high-energy photons has emerged in the early years of the 21st century, and it has been widely adopted for the management of LAPC, since it results in significantly less gastrointestinal toxicity compared to three-dimensional conformal RT (3D-CRT) and may enable dose escalation [10]. In addition, SBRT was found to significantly improve the 2-year OS compared with CFRT, with lower rates of acute grade 3/4 toxicity and no difference in late toxicity [11]. Physical and biological properties of PBT decrease the radiation dose to the surrounding organs at risk (OAR), possibly allowing for dose escalation [12]. However, factors such as the optimal radiation total dose, the fractionation schedule and the radiation technique remain unknown [13].

In order to find evidence for the best radiation technique, radiation modality and fractionation schedule in LAPC, in this systematic review, we summarize the treatment outcomes and acute as well as long-term toxicity of both photon and proton radiotherapy with normo- and hypo-fractionated schedules in patients with locally advanced unresectable pancreatic adenocarcinoma.

## 2. Materials and Methods 

The systematic review was based on PubMed, MEDLINE (via PubMed) and Cochrane databases to find studies published between 2012 and 2022, meeting our inclusion criteria (see below). Relevant studies were included according to the Population, Intervention, Control, Outcome, and Study Design (PICOS) selection protocol. Furthermore, the systematic review followed the recommendations of the Preferred Reporting Items for Systematic Reviews and Meta-Analyses (PRISMA) (Figure 1). The protocol has not been registered. 

### 2.1. Search Strategy

The search strategies were developed using PubMed Mesh terms using the same terms with different Boolean operators as follows: (“pancreatic cancer” OR “pancreas cancer” OR “pancreas neoplasm” OR “cancer of the pancreas”) AND (“locally advanced” OR “unresectable”) AND (“Proton Therapy” OR “Proton Beam Therapy” OR “Proton Beam Radiation Therapy” OR “Radiation therapy” OR “Radiotherapy” OR “Radiation treatment” OR “targeted radiotherapy” OR “targeted radiation therapy”).

### 2.2. Inclusion and Exclusion Criteria

The inclusion criteria were: (1) pathologically proven locally advanced unresectable cN0-N1 cM0 pancreatic adenocarcinoma; (2) patients aged 18 years or older; (3) single or multi-arm phase 2/3 prospective studies for normo- or hypo-fractionated photon therapy combined with chemotherapy; (4) prospective single or multi-arm studies for normo- or hypo-fractionated proton therapy; (5) retrospective studies; (6) reporting of at least one toxicity or one survival parameter; and (7) studies published between 2012 and 2022. Studies with mixed populations were included only if the outcomes were reported separately for each patient cohort. Exclusion criteria were conference abstracts, phase I studies for photon therapy, duplicates, systematic reviews, meta-analyses, and languages other than English. Reviewers (Sally A. Elkhamisy and Esther G. C. Troost) rated all identified studies according to the inclusion and exclusion criteria. In case of discordance, consensus was sought.

### 2.3. Data Extraction and Statistical Analysis

Data extracted from the screened studies included population characteristics, type of study, type of radiation beam therapy, radiotherapy total dose and dose per fraction, radiotherapy technique, chemotherapy applied, acute and late toxicities and survival outcomes. Statistical analyses were performed using Microsoft Excel for Microsoft Office 365 (version 2301) for descriptive statistics and GraphPad Prism software version 6.03 for Windows (GraphPad Software, San Diego, CA, USA).

### 2.4. Bias Assessment

The risk of bias was evaluated using risk of bias tools offered by Cochrane and other tools established in other systematic reviews, adapted and defined according to our review [14,15,16] (see Appendix A). 

## 3. Results

### 3.1. Search Results

Based on the search strategy, 1111 studies published between 2012 and 2022 were identified. After duplicate removal and application of the eligibility criteria, a total of 30 studies were assessed in a full text review. An additional three studies were excluded since no full text could be retrieved. Moreover, four studies included in previous systematic reviews, which seemed not to be involved in the search results, but that nonetheless met the inclusion criteria, were added. Thus, a total of 31 studies were included in the systematic review. Table 1, Table 2 and Table 3 provide an overview of treatment characteristics and outcome measures, such as radiotherapy and chemotherapy protocols, as well as acute and late toxicities, local control and overall survival. 

### 3.2. Studies and Patient Characteristics 

The 31 studies were classified into two groups: (1) a photon therapy group, which was further split into studies including (a) CFRT and (b) SBRT, and (2) a PBT group. The total number of studies in the CFRT group was 15, of which 12 were phase II [17,18,20,21,23,24,25,26,27,28,29,30], one was phase III [19], and two were retrospective studies [22,31]. In total, ten studies were allocated to the SBRT group, of which six were phase II [32,33,35,36,37,38] and four were retrospective studies [34,39,40,41]. For the PBT group, six studies were identified: four prospective [43,44,45,47] and two retrospective studies [42,46].

Regarding the total number of patients included, we considered only the number of patients with LAPC who received chemoradiation, not the whole study population. The systematic review included a total number of 1659 patients, of which 971 patients were treated in the CFRT group, 401 in the SBRT group, and 287 in the PBT group.

### 3.3. Radiation Therapy

The range of median total radiation dose reported in the CFRT group was 50.4–59.4 Gy, with the median number of fractions in the range of 25–33 fractions. Three studies administered a median total dose of 54 Gy over 30 fractions [19,20,27]. However, five studies delivered a median dose of 50.4 Gy over 28 fractions [23,25,28,29,30]. In two studies, a total dose of 45 Gy in 25 fractions was delivered to a larger target volume with a simultaneous integrated boost (SIB) to boost the volume up to 54 Gy [17,21].

In one study, a total dose of 44 Gy was the base plan and SIB up to 55 Gy was the boost plan adopted [31], while in Esnaola et al. [18] the total dose was 45.9 Gy/30 fractions with SIB of 54 Gy. Youl et al. [22] administered a median total dose of 52.5 Gy in 30 fractions with dose per fraction of 1.75 Gy. One study applied the total radiation dose of 59.4 Gy in 33 fractions [26]. The most common radiotherapy technique used was 3DCRT (seven studies) [19,20,26,27,28,29,30], whereas IMRT was adopted in five studies [17,18,21,25,31]. Two studies reported on using either 3DCRT or IMRT [22,23]. 

In the SBRT group, it was noted that the range of cumulative radiotherapy dose was 24 Gy to 50 Gy. The calculated median biological effective dose (BED10) ranged between 37.5 Gy and 85.5 Gy. The highest dose per fraction was reported by Quan et al. [32] (12 Gy per fraction to a total dose of 36 Gy) and the lowest dose per fraction was reported by Moningi et al. [34] (5 Gy per fraction of a total dose of 25 Gy) and Teriaca et al. [37] (5 Gy per fraction for a total dose of 40 Gy). The total number of fractions across the studies ranged from three to eight fractions. 

Regarding PBT, two studies delivered a total dose of 67.5 Gy (RBE) in 25 fractions with 2.7 Gy (RBE) per fraction [43,46]. One study prescribed 45 Gy (RBE) or 50 Gy (RBE) and 30 Gy (RBE) to the PTVs in 10 fractions using a SIB technique [42]. Two studies administered 59.4 Gy (RBE) in 33 fractions with 1.8 Gy (RBE) per fraction [44,45]. Maemura et al. [47] prescribed a total dose of 50 Gy (RBE) or an escalated dose of 67.5 Gy (RBE), both in 25 fractions, if dose escalation appeared to be feasible. Of note, the radiation technique employed throughout the studies was clearly reported as passive scattered proton therapy in two studies [42,45].

### 3.4. Chemotherapy Regimens 

The chemotherapy protocols reported were very heterogenous across the studies. For the CFRT and PBT groups, the most commonly reported concurrent chemotherapy regimens were fluoropyrimidine-based chemotherapy (using either capecitabine or S1) or gemcitabine-based chemotherapy. Concurrent chemotherapy protocols using cisplatin/docetaxel combination was also reported in Ducreux et al. [20]. Chemotherapy/target therapy combinations were investigated in recent years. Concurrent gemcitabine and cetuximab was administered in two studies [17,18]; whereas, in Himmel et al. [19], erlotinib was given as an induction or maintenance therapy combined with gemcitabine in the chemoradiation arm.

In general, for SBRT, no chemotherapy was given concurrently, but was neoadjuvant or adjuvant instead. Only Lin et al.’s [41] patients were treated with either gemcitabine or FOLFOX concurrently with SBRT.

### 3.5. Acute G3/G4 or Higher Toxicity

The Common Terminology Criteria of Adverse Events (CTCAE) or equivalent was used for toxicity evaluation. However, in four studies, the toxicity evaluation scales were not reported [21,22,34,45]. 

In the CFRT group, acute G3/G4 toxicity was not observed in one study [21], not reported in two studies [22,24] and shown as combined results with other cohorts of patients such as BRPC in two studies [18,26]. Total hematological and non-hematological toxicities were reported in four studies [19,20,23,31], with the highest acute toxicity reported in Ducreux et al. [20] (51%) and the lowest acute toxicity reported in the capecitabine arm in Mukherjee et al. [23] (12%). The remaining studies reported the acute toxicity individually for each side effect as listed in Table 1.

For the SBRT group, seven studies out of ten did not observe acute G3 or higher toxicities [32,35,36,37,39,40,41]. In the remaining three studies, the toxicity was reported as follows: one study reported combined results with the other cohort of patients in the study [34], Zhu et al. [33] showed 14% total G3 toxicity, and in Herman et al. [38] the percentage of G3 or more gastrointestinal toxicity was 2%.

Regarding PBT, G3/G4 toxicity was not observed in three studies [42,44,45]. On the contrary, Terashima et al. [43] and Ogura et al. [46] reported G3/G4 hematological toxicity of 73.5% and 44%, respectively, while the non-hematological toxicity (GIT) was 21% and 0%, respectively. Moreover, Maemura et al. [47] reported that only one patient developed a G3 gastric ulcer.

### 3.6. Late G3/4 or Higher Toxicity

In the CFRT group, late toxicity was not reported in 13 out of 15 studies. Liermann et al. [17] did not observe G3/G4 late toxicity in arm A; however, in arm B a G3 ileus (3%) and a G3 GIT hemorrhage (1%) were reported. Furthermore, Oh et al. [31] reported G3 anemia (2%) without any other G3 hematological or GIT toxicity.

For the SBRT group, G3/G4 toxicities were not observed in two studies [35,36], not reported in two studies [32,41] and combined with other cohorts of patients in one study [34]. In three studies, the GI toxicities ranged from 1.7% to 10% [37,38,40], while in Zhu et al. [33] and Chong et al. [39], total G3 toxicity was 4.8% and 5.3%, respectively. 

In the PBT group, grade 3 or higher late toxicities were not reported in three studies [44,45,47] and not observed in one study [42]. Conversely, Terashima et al. [43] and Ogura et al. [46] published non-hematological grade 3 or higher toxicities of 17% and 9%, respectively.

### 3.7. Survival Outcomes

Median OS analysis of all groups showed that most of the studies in both CFRT and SBRT reported comparable values (average 15 months), while the PBT group showed more prolonged OS (average 20 months; see Figure 2). 

The median OS reported in the CFRT group ranged from 9.3 to 22.9 months, while the 1-year OS rate was 36–88.9% (reported in ten studies [17,20,21,22,23,25,27,28,29,30]) and the 2-year OS rate was 8–43.5% (reported in eight studies [17,21,22,25,27,28,29,30]).

The SBRT group showed a range of median OS of 8.5–20 months, with a 1-year OS rate of 53.9–80% (reported in eight studies [32,33,36,37,38,39,40,41]) and a 2-year OS rate of 16–35.1% [32,36,38,40]. Meanwhile, in the PBT group, the range of median OS was 18.4–22.3 months, the 1-year OS rate was 61–80% (reported in four studies [43,45,46,47]) and the 2-year OS rate was 31–50% (reported in three studies [45,46,47]), respectively.

### 3.8. Overall Risk of Bias

Analysis of the risk of bias for all studies was performed and reviewed by two reviewers independently. For the CFRT group, five studies were evaluated to be at high risk according to the adopted domains, while ten studies were evaluated to be low risk (Table 4). Five studies in the SBRT group were rated to be at high risk and five studies were rated as low risk (Table 5). Furthermore, in the proton therapy group, three studies were considered to be high risk and three studies were low risk (Table 6).

## 4. Discussion

In this systematic review, different radiation modalities (i.e., photons and protons) as well as fractionation schedules (normo- and hypo-fractionation) applied in the treatment of LAPC were systematically reviewed regarding toxicity profile and treatment outcome.

Patients treated with SBRT were found to have less acute toxicity compared to those treated with CFRT (Table 1 and Table 2). However, in the PBT group, it was observed that patients treated with high doses (67.5 Gy (RBE)) or more experienced high G3 acute and late toxicity [43,46]. Regarding the survival, median OS and 1-year OS showed comparable values for CFRT and SBRT, while PBT showed a promising median OS, with a higher 1-year OS rate. Since most studies did not report either late toxicity or 2-year OS, we were unable to compare these parameters between the groups. Interestingly, a recent systematic review and meta-analysis with different selection criteria compared to our study showed no statistically significant difference in 1-year OS between SBRT and CFRT (53.7% vs. 49.3%, *p* = 0.63), however, a significant improvement in 2-year OS was noted for SBRT (26.9% vs. 13.7%, *p* = 0.004), with lower rates of acute grade 3/4 toxicity (5.6% vs. 37.7%) and no difference in late toxicity [11]. 

SBRT is based on the delivery of high radiation doses per fraction in a small number of fractions (≤5 fractions) ascertaining high precision and accuracy with minimal setup errors [48]. Compared to CFRT, SBRT is delivered over a shorter overall treatment time, and less acute toxicity is expected [6]. Nevertheless, the total dose and fractionation schedule of SBRT are still a matter of debate, since early clinical studies using single fractions of 25 Gy, or three fractions of 15 Gy reported unacceptably high GI toxicities [49,50]. Therefore, somewhat lower radiation doses per fraction and more advanced SBRT techniques, including motion management, have been adopted in recent clinical trials. Most likely, these technological advancements will even improve treatment outcomes in terms of local control, possibly survival and toxicity.

Furthermore, the utilization of particle therapy, specifically proton beam therapy (PBT), to treat LAPC has shown potential as a radiotherapy technique. This is due to the unique physical properties of PBT that allow for full dose delivery to a specific depth (Bragg peak) with minimal exposure to surrounding tissues, in particular at the distal end. This may facilitate the delivery of high doses to the tumor and minimize the dose received by healthy structures, ultimately widening the therapeutic window [51]. Dealing with abdominal malignancies is particularly challenging for PBT, since it is more sensitive to density changes (e.g., stomach and bowel filling) compared to photon-based radiation therapy. Changes in the path of the proton beam due to organ filling or movement can change the position of the Bragg peak relative to the target volume [52]. Specific protocols for patient preparation (e.g., fasting prior to treatment), pre-treatment imaging as well as motion mitigation strategies (e.g., abdominal compression corset [8], re-scanning, log-file based treatment delivery) may increase the usage of proton beam therapy for LAPC and thus propel the generation of clinical evidence. 

To date, due to the limited availability of proton therapy and its associated costs, most of the available proton studies for LAPC are non-randomized studies, recruited a small number of patients, or were designed as retrospective studies. The range of total dose used across the studies was 50–67.5 Gy (RBE), with promising local control and survival rates, but still the toxicity profile—especially with high doses of 67.5 Gy (RBE) or higher—is a matter of concern [42,43,45]. More well-designed, technologically advanced clinical studies are required to provide evidence on the value of PBT as opposed to photon therapy in the management of LAPC.

Moreover, the evolution of magnetic resonance imaging-guided radiotherapy (MRIgRT) using MR-hybrid machines is considered a promising technique that provides both anatomical and functional information on a daily basis. Moreover, since online plan re-optimization is feasible and target volumes can be monitored in real-time, MRIgRT may allow for smaller target volumes to be treated and thus OAR to be spared [53,54]. A clinical study using MRIgRT for unresectable pancreatic carcinoma is currently being recruited (NCT01972919).

Noteworthy in our systematic review, is that different chemotherapy protocols were used across the studies. Since assessment of the efficacy of chemotherapy was not the aim of this systematic review, we will not conclude on the superiority of a specific protocol. However, a meta-analysis published in 2016 assessed the benefit of FOLFIRINOX in LAPC and concluded that FOLFIRINOX was associated with a high median OS (24 months). Moreover, a promisingly high number of patients (63.5%) received subsequent radio-chemotherapy, and 25.9% underwent surgical tumor resection [55]. Induction chemotherapy before radio-chemotherapy may be considered as an option in selected non-metastatic patients with good performance status [56]. This approach may improve systemic disease control and, moreover, help in selecting patients benefitting from radio-chemotherapy based on their response to induction chemotherapy. Although FOLFIRINOX showed an interesting response regarding tumor respectability in such patients, there is no clear consensus on the optimal chemotherapy regimen to date [57].

This systematic review is subjected to several limitations. The studies extracted are mixed phase II/III studies or retrospective studies with different inclusion and exclusion criteria, which result in biases as reported. Moreover, the wide variation in total radiation doses and doses per fraction, particularly in those studies concentrating on SBRT and PBT, along with incomplete reporting of data, prohibits the direct comparison of the toxicity and outcome data of the three modalities. Even though there seems to be a direct correlation between the incidence of toxicity (acute or late) and the fractionation schedule and radiation modality, it was not possible to address this by statistical testing, since substantial amounts of data were missing and different toxicity scoring metrices were used. Finally, the number of studies reporting on PBT was limited, as was the number of patients included. Thus, the role of SBRT and PBT in the treatment of LAPC is still to be established.

## 5. Conclusions

In conclusion, this systematic review revealed comparable median and 1-year overall survival for CFRT and SBRT, with a more favorable toxicity profile of SBRT. PBT is a promising new treatment modality for LAPC patients, however, further clinical studies, ideally using different fractionation schedules and including photon regimen as comparator, are needed.

## Figures and Tables

**Figure 1 cancers-15-03771-f001:**
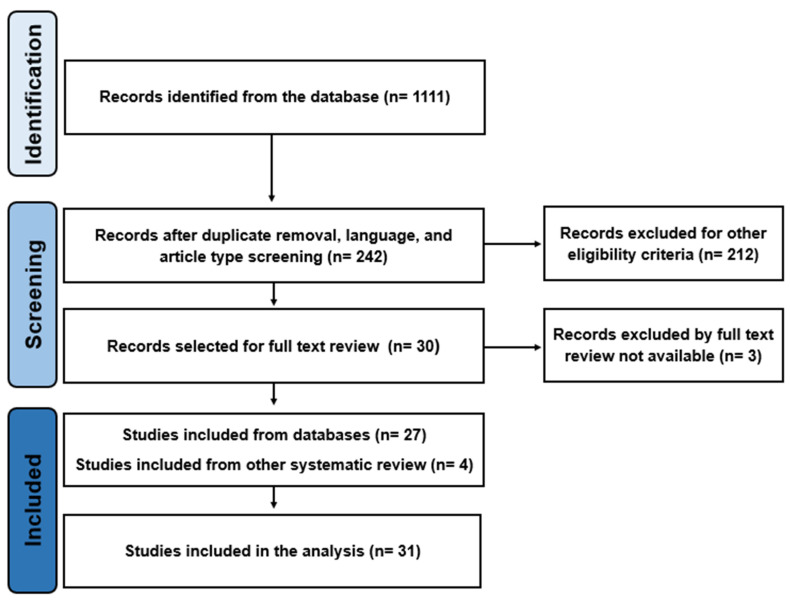
PRISMA flow chart illustrating the screening and the selection process.

**Figure 2 cancers-15-03771-f002:**
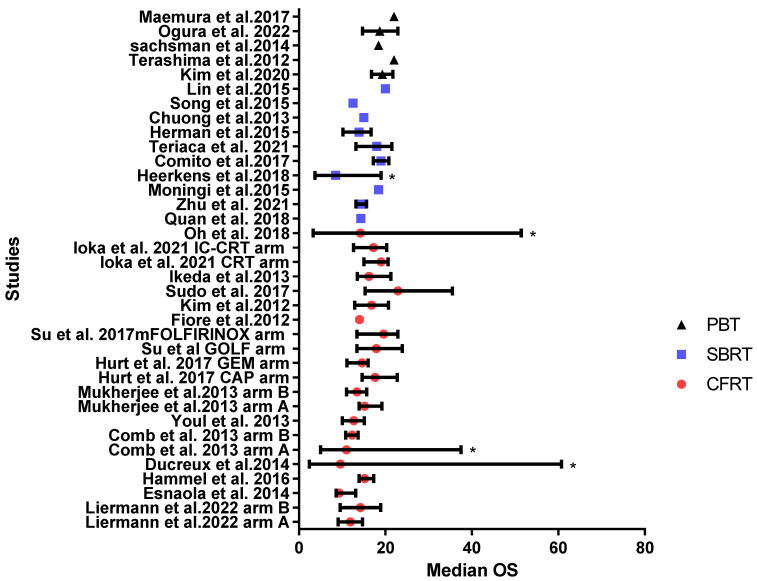
Reported median OS for patients with LAPC across the CFRT group (
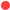
), SBRT (
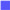
), and proton beam therapy groups (
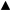
). Bars represent 95% CI or range (range marked by *) [17,18,19,20,21,22,23,24,25,26,27,28,29,30,31,32,33,34,35,36,37,38,39,40,41,42,43,45,46,47].

**Table 1 cancers-15-03771-t001:** Clinical studies on conventionally fractionated radiotherapy (CFRT) for LAPC.

Study	Study Type	N	Dose/Gy	Technique	Chemotherapy	Median OS, Months	1-Year OS Rate, %	2-Year OS Rate, %	Local Control	Acute G3/4 Toxicity	Late G3/4 Toxicity
Liermann et al., 2022, arm A [17]	(Long-term results of phase II PARC trial)	35	45 Gy/25 fractions with SIB to 54 Gy	IMRT	Concurrent gemcitabine/cetuximab followed by maintenance gemcitabine	11.9	47%	14.7%	1-year LC rate approximately 65% (from figure)	G3 nausea: 4% G3/4 leucopenia: 37% G3/4 anemia: 14%	Not observed
Liermann et al., 2022, arm B [17]	(Long-term results of phase II PARC trial)	33	45 Gy/25 fractions with SIB to 54 Gy	IMRT	Concurrent gemcitabine/cetuximab followed by maintenance gemcitabine/cetuximab	14.2	60.6%	27%	1-year LC rate approximately 85% (from figure)	G3 nausea: 9% G3/4 leucopenia: 25% G3/4 anemia: 9%	G3 ileus: 3% G3 GIT Hge: 1%
Esnaola et al., 2014 [18]	Phase II	19	45.9 Gy/30 fractions with SIB to 54 Gy	IMRT	Induction gemcitabine/oxaliplatin /cetuximab + concurrent capecitabine	9.3	NR	NR	8.3% underwent R0 resection	Combined with BRPC patients	NR
Hammel et al., 2016, CRT arm [19]	Phase III	133	54 Gy/30 fractions	3DCRT	Concurrent capecitabine, either induction gemcitabine alone or gemcitabine + erlotinib followed by maintenance therapy	15.2	NR	NR	Loco-regional progression rate: 32%	Total hematological toxicity: 3.9% Total non-hematological toxicity: 23.1%	NR
Ducreux et al., 2014 [20]	Phase II	51	Median dose 54 Gy/30 fractions	3DCRT	Concurrent docetaxel/cisplatin	9.6	41%	NR	CR + PR: 27% SD: 51% PD: 14% (8% not evaluated)	GIT toxicity: 43% hematological toxicity: 8%	NR
Comb et al., 2013, arm A [21]	Phase II	57	A median total dose of 45 Gy/25 fractions with SIB to 54 Gy	IMRT	Concurrent and adjuvant gemcitabine	11	36%	8%	SD: 88% PD: 5% PR: 7%	Not observed	NR
Comb et al., 2013, arm B [21]		198	Median dose of 52.2 Gy/28 fractions	3DCRT	Concurrent and adjuvant gemcitabine	12.3	NR	NR	SD: 80% PD: 11% PR: 9%	Not observed	NR
Youl et al., 2013 [22]	Retrospective	74	Median total dose of 52.5 Gy/30 fractions	3DCRT or IMRT	Induction and concurrent gemcitabine	12.7	53.6%	14.7%	SD: 60.8% PD: 22.9%	NR	NR
Mukherjee et al., 2013, capecitabine arm (SCALOP trial) [23]	Phase II	36	Median total dose of 50.4 Gy/28 fractions	3DCRT or IMRT	Induction gemcitabine/capecitabine + concurrent capecitabine	15.2	79.2%	NR	CR: 6% PR: 17% SD: 63% PD: 14%	Both hematological and non-hematological: 12%	NR
Mukherjee et al., 2013, gemcitabine arm (SCALOP trial) [23]	Phase II	38	Median total dose of 50.4 Gy/28 fractions	3DCRT or IMRT	Induction gemcitabine/capecitabine + concurrent gemcitabine	13.4	64.2%	NR	CR: 0% PR: 19% SD: 67% PD: 14%	Both hematological and non-hematological: 37%	NR
Hurt et al., 2017 [24]	(Long-term results of SCALOP trial)					Capecitabine arm: 17.6 Gemcitabine arm: 14.6					
Su et al., 2022, GOLF arm [25]	Phase II	17	50.4 Gy/28 fractions	IMRT	Induction GOLF followed by concurrent gemcitabine	17.9	82.1%	31.8%	PR: 24% SD: 65% PD: 11%	G3/4 leukopenia: 29.4%	NR
Su et al., 2022, mFOLFIRINOX arm [25]	Phase II	21	50.4 Gy/28 fractions	IMRT	Induction m FOLFIRINOX followed by concurrent 5-FU	19.6	88.9%	29.6%	PR: 29%, SD: 33% PD: 38%	G3/4 leukopenia: 0%	NR
Fiore et al., 2017 [26]	Phase II	27	59.4 Gy/33 fractions	3DCRT	IC gemcitabine/oxaliplatin then concurrent gemcitabine	14	Combined with BRPC patients	Combined with BRPC patients	40.7% became resectable	Combined with BRPC patients	NR
Kim et al., 2012 [27]	Phase II	25	Median total dose of 54 Gy/30 fractions	3DCRT	Induction gemcitabine/cisplatin—concurrent capecitabine followed by gemcitabine	16.8	69.1%	16.1%	PD: 14.3%	G3 neutropenia: 4% G3 anemia: 4% G3 nausea/vomiting: 4% G3 diarrhea: 8%	NR
Sudo et al., 2017 [28]	Phase II	23	50.4 Gy/28 fractions	3DCRT	Induction gemcitabine/S1 then CRT with S1 followed by maintenance therapy with S1	22.9	82.6%	43.5%	Significant reduction in tumor size (median size, from 41 to 32 mm)	G3 neutropenia: 4.3% G3 biliary tract infection: 8.7%	NR
Ikeda et al., 2013 [29]	Phase II	53	50.4 Gy/28 fractions	3DCRT	Concurrent S1 followed by maintenance S1	16.2	72%	26%	PR: 27%, SD: 67%	G3 leukocytopenia: 10% G3 gastric ulcer: 2%	NR
Ioka et al., 2021, CRT arm [30]	Phase II	50	50.4 Gy/28 fractions	3DCRT	Concurrent S1 + maintenance gemcitabine	19	66.7%	36.9%,	NR	G3 leucopenia: 60% G3 anorexia: 16% G3 biliary infection: 18%	NR
Ioka et al., 2021, IC-CRT arm [30]	Phase II	34	50.4 Gy/28 fractions	3DCRT	Induction gemcitabine then CRT + maintenance gemcitabine	17.2	69.3%	18.9%	NR	G3 leucopenia: 59% G3 anorexia: 4% G3 biliary infection: 27%	NR
Oh et al., 2018 [31]	Retrospective	47	44 Gy/22 fractions with SIB to 55 Gy	IMRT	Concurrent chemotherapy, with gemcitabine (*n* = 37) and capecitabine (*n* = 10) with or without induction chemotherapy	14.2	NR	NR	PR: 61.7% SD: 38.3%	G3 hematological toxicity: 12.8% G3 GIT toxicity: 2.1%	G3 anemia: 2.1%

OS: overall survival, 3DCRT: three-dimensional conformal radiotherapy, IMRT: intensity-modulated radiotherapy, Hge: hemorrhage, LC: local control, SIB: simultaneous integrated boost, BRPC: borderline resectable pancreatic cancer, Gy: Gray, CR: complete response, PD: progressive disease, PR: partial response, SD: stable disease, GIT: gastrointestinal tract, NR: not reported, CRT: chemoradiotherapy, IC: induction chemotherapy, GOLF: gemcitabine, oxaliplatin, leucovorin, 5-fluorouracil, FOLFIRINOX: leucovorin calcium, fluorouracil, oxaliplatin, irinotecan, 5 FU: 5-fluorouracil.

**Table 2 cancers-15-03771-t002:** Clinical studies of stereotactic body radiotherapy (SBRT) for LAPC.

Study	Study Type	N	Total Dose/Gy	Chemotherapy	Median OS, Months	1-Year OS Rate, %	2-Year OS Rate, %	Local Control	Acute G3/4 Toxicity	Late Toxicity
Quan et al., 2018 [32]	Phase II	16	36 Gy/3 fractions	Induction gemcitabine/capecitabine ± adjuvant chemotherapy	14.3	60%	16%	m LPFS: 28.1 months 1-year LPFS rate: 78% 2-year LPFS rate: 52%	Not observed	NR
Zhu et al., 2021 [33]	Phase II	63	35–40 Gy/5 fractions	Sequential S1 one month after SBRT	14.4	73%	NR	Local recurrence rate: 15.9% regional recurrences: 30.2%	Total G3 toxicity: 14.3%	Total G3 toxicity: 4.8%
Moningi et al., 2015 [34]	Retrospective	Unresectable: 74	25–33 Gy/5 fractions	Different IC regimens followed by post SBRT chemotherapy	18.4	Reported combined with BRPC patients	Reported combined with BRPC patients	15 patients underwent surgery, LPFS combined with BRPC patients	Reported combined with BRPC patients	reported combined with BRPC patients
Heerkens et al., 2018 [35]	Phase II	20	24 Gy/3 fractions	±Post SBRT gemcitabine/nab-paclitaxel or FOLFIRINOX	8.5	NR	NR	SD: 39% PD: 61% CR or PR: 0%	Not observed	Not observed
Comito et al., 2017 [36]	Phase II	45	45 Gy/6 fractions	Pre or post SBRT different chemotherapy regimens	19 months from diagnosis	85% from diagnosis	33% from diagnosis	m FFLP: 26 months	Not observed	Not observed
Teriaca et al., 2021 [37]	Phase II	39	40 Gy/8 fractions	IC FOLFIRINOX	18	77%	NR	1-year LC rate: 80%, 3-year LC rate: 53%	Not observed	G3 toxicity or higher: 10%
Herman et al., 2015 [38]	Phase II	49	33 Gy/5 fractions	Induction and post SBRT Gemcitabine	13.9	59%	18%	1-year FFLP rate: 78%	G3 or more gastric ulcer: 2% G3 lymphopenia: 8.2%	G3 or more gastric ulcer: 6.4%
Chuong et al., 2013 [39]	Retrospective	16	35–50 Gy/5 fractions	Induction gemcitabine followed by SBRT	15	68.1%	NR	Surgical resection: 12.5% (other LC parameters were combined with BRPC)	Not observed	Total G3 toxicity: 5.3%
Song et al., 2015 [40]	Retrospective	59	Median total dose of 45 Gy/5 fractions	90% of patients received chemotherapy before or after treatment	12.5	53.9%	35.1%	1-year FFLP rate was 90.8%	Not observed	G3 GIT reaction: 1 patient
Lin et al., 2015, SBRT arm [41]	Retrospective	20	35–45 Gy/5 fractions	Different concurrent chemotherapy regimens	20	80%	NR	1-year LDFS approximately 70% (from the figure)	Not observed	NR

m LPFS: median local progression-free survival, m FFLP: median freedom from local progression, LDFS: local disease-free survival, NR: not reported.

**Table 3 cancers-15-03771-t003:** Clinical studies of proton beam therapy for LAPC.

Study	Study Type	N Patients	Total Dose Gy (RBE)	Technique	Chemotherapy	Median OS, Months	1-Year OS Rate, %	2-Year OS Rate, %	Local Control	Acute G3/4 Toxicity	Late G3/4 Toxicity
Kim et al., 2020 [42]	Retrospective	81	PTV1: 45 or 50 Gy PTV2: 30 Gy/10 fractions	Passive scattered proton beams	Upfront or maintenance chemotherapy or neither ± concurrent fluoropyrimidine chemotherapy	19.3	NR	NR	PR: 6.% SD: 80% PD: 14%	Not observed	Not observed
Terashima et al., 2012 [43]	Phase I/II	5	50 Gy/25 fractions	NR	Concurrent and adjuvant gemcitabine	NR	NR	NR	NR	G3 leukopenia: 20% G3 neutropenia: 20% G3 fatigue: 20% G3 anorexia: 20%	NR
Terashima et al., 2012 [43]	Phase I/II	5	70.2 Gy/26 fractions	NR	Concurrent and adjuvant gemcitabine	NR	NR	NR	NR	G3 leukopenia: 60% G3 neutropenia: 40% G3 anorexia: 20%	G3 GIT: 20%
Terashima et al., 2012 [43]	Phase I/II	40	67.5 Gy/25 fractions	NR	Concurrent and adjuvant gemcitabine	NR	78.8%	NR	1-year FFLP: 79.9%	G3 hematological toxicity: 66% G4: 7.5%, G3 GIT toxicity: 21%	G3 anorexia: 3%, G3 gastric ulcer: 8% G3 fatigue: 3% G5 gastric ulcer: 3%
Terashima et al., 2012 [43]	Phase I/II	50	50–70.2 Gy/ 25–26	NR	Concurrent and adjuvant gemcitabine	22 (from the curve)	76.8%	NR	1-year FFLP: 81.7%		
Nichols et al., 2013 [44]	Prospective single institute	Unresectable: 12	Medan dose 59.4 Gy/33 fractions	NR	Concurrent capecitabine	8.8	NR	NR	NR	Not observed	NR
Sachsman et al., 2014 [45]	Prospective single institute	11	59.4 Gy/33 fractions	Passive scattering proton beams	Concomitant capecitabine	18.4	61%	31%	1- and 2-year FFLP: 86% and 6%	Not observed	NR
Ogura et al., 2022 [46]	Retrospective	123	67.5 Gy/25 fractions	NR	Concurrent and adjuvant gemcitabine	18.7	70.4%	35.7%	1-year LPFS: 78.2% 2-year LPFS: 59%	All are hematological: G3: 42% G4: 2%	GIT toxicity: G3: 5% G4: 2% G5: 2%
Maemura et al., 2017, PT arm [47]	Prospective non-randomized	10	50 Gy or 67.5 Gy/25 fractions	NR	IC followed by concurrent S1 + maintenance S1	22.3	80%	45%	PR: 20% SD: 60% PD: 20%	One patient developed G3 ulcer	NR

RBE: relative biological effectiveness, NR: not reported.

**Table 4 cancers-15-03771-t004:** Risk of bias assessment for the CFRT group.

Study	Major Domains	Minor Domains	
Selection of Participants	Clarification of Intervention	Measurements of Outcomes	Missing Data	Funding	Conflict of Interest	Overall
Liermann et al., 2022 [17]							
Esnaola et al., 2014 [18]							
Hammel et al., 2016 [19]							
Ducreux et al., 2014 [20]							
Comb et al., 2013 [21]							
Youl et al., 2013 [22]							
Mukherjee et al., 2013 [23]							
Hurt et al., 2017 [24]							
Su et al., 2022 [25]							
Fiore et al., 2017 [26]							
Kim et al., 2012 [27]							
Sudo et al., 2017 [28]							
Ikeda et al., 2013 [29]							
Ioka et al., 2021 [30]							
Oh et al., 2018 [31]							


: Low risk; 

: High risk; 

: Unclear risk.

**Table 5 cancers-15-03771-t005:** Risk of bias assessment for the SBRT group.

Study	Major Domains	Minor Domains	
Selection of Participants	Clarification of Intervention	Measurements of Outcomes	Missing Data	Funding	Conflict of Interest	Overall
Quan et al., 2018 [32]							
Zhu et al., 2021 [33]							
Moningi et al., 2015 [34]							
Heerkens et al., 2018 [35]							
Comito et al., 2017 [36]							
Teriaca et al., 2021 [37]							
Herman et al., 2015 [38]							
Chuong et al., 2013 [39]							
Song et al., 2015 [40]							
Lin et al., 2015 [41]							


: Low risk; 

: High risk; 

: Unclear risk.

**Table 6 cancers-15-03771-t006:** Risk of bias assessment for the proton therapy group.

Study	Major Domains	Minor Domains	
Selection of Participants	Clarification of Intervention	Measurements of Outcomes	Missing Data	Funding	Conflict of Interest	Overall
Kim et al., 2020 [42]							
Terashima et al., 2012 [43]							
Nicolas et al., 2013 [44]							
Sachsman et al., 2014 [45]							
Ogura et al., 2022 [46]							
Maemura et al., 2017 [47]							


: Low risk; 

: High risk; 

: Unclear risk.

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
