# Peer review of "Normo- or Hypo-Fractionated Photon or Proton Radiotherapy in the Management of Locally Advanced Unresectable Pancreatic Cancer: A Systematic Review"

_cancers, 2023, doi:10.3390/cancers15153771_

Round 1
Reviewer 1 Report
I would like to congratulate the authors to this comprehensive review of the literature and excellent presentation of the study results.
Introduction: In the introduction it is mentioned that the standard of care is introduction chemotherapy, especially with FOLFIRINOX, and paclitaxel and gemcitabine. Here, I missed survival data achieved with these standard of care treatments (FOLFIRINOX versus Gemcitabine-based regimens).
Results:
Page 10, line 7-8: For the PBT group, five studies were identified: four prospective [42]–[45], and two retrospective studies. This sentence is contradictory: 5 studies: 4 + 2 = 6.
Page 12, Figure 2: results of 5 studies with proton therapy are depicted. However, in the Table 3: there are 6 studies with median OS data. Nichols et al 2013 with an OS of 8.8 months is omitted in the Figure 2. Why?
Discussion:
Page 1, line 10: it is written that results for PBT are promising. It is noteworthy to mention in the discussion that the PBT studies presenting OS data are either very small prospective single institution trials with 10 patients or retrospective trials. This group also mixes stereotactic and normofractionated RT studies.
Page 2, line 57 - 63: Although it was not the scope of this trial to assess different chemotherapy regimens, it is important to emphasize more the importance of the chemoregimen FOLFIRINOX, how radiotherapy can be integrated into it, which role surgery might play and how future studies might look like.
Page 2, line 70 and 71: did you mean induce instead of introduce?
Author Response
Introduction
Point 1:
In the introduction it is mentioned that the standard of care is introduction chemotherapy, especially with FOLFIRINOX, and paclitaxel and gemcitabine. Here, I missed survival data achieved with these standard of care treatments (FOLFIRINOX versus Gemcitabine-based regimens).
Response 1:
FOLFIRINOX was evaluated in the PRODIGE 4-ACCORD11 trial, comparing standard dose gemcitabine to FOLFIRINOX, but only metastatic patients were included in the trial. This trial showed a median overall survival benefit in favor of FOLFIRINOX. In another phase 3 trial (MPACT), metastatic patients were randomized to either (Gem-nabP) or gemcitabine alone and both median overall survival and progression-free survival were superior in the Gem-nabP group.
Until now, no randomized controlled trial comparing FOLFIRINOX with Gemcitabine in patients with locally advanced pancreatic cancer has been performed. Thus, the NCCN guidelines recommendations for FOLFIRINOX or modified FOLFIRINOX and gemcitabine + albumin-bound paclitaxel in patients with locally advanced disease are based on extrapolations from randomized trials in patients with metastatic disease. Noteworthy, a meta-analysis published in 2016 assessed the benefit of FOLFIRINOX in LAPC and concluded that FOLFIRINOX was associated with a high median overall survival (24 months), as is mentioned in the discussion of the manuscript. (page 2 -lines 60-62)
Point 2:
Page 2, line 70 and 71: did you mean induce instead of “introduce”?
Response 6: Thank you for the correction. The word “introduced” was replaced by the word “induced”. (Page 2, Line 71-72).
Results
Point 3:
Page 10, line 7-8: For the PBT group, five studies were identified: four prospective [42]–[45], and two retrospective studies. This sentence is contradictory: 5 studies: 4 + 2 = 6.
Response 2:
Indeed, the phrase was incorrect. It has been revised and corrected accordingly (page 11, line 7-8).
Point 4:
Page 12, Figure 2: results of 5 studies with proton therapy are depicted. However, in Table 3: there are 6 studies with median OS data. Nichols et al 2013 with an OS of 8.8 months is omitted in Figure 2. Why?
Response 3: Thank you for pointing this out. Since the median survival time reported by Nichols et al. included patients with ampullary tumor, and the data could not be separated, we did not include those data in Figure 2.
Discussion
Point 5:
Page 1, line 10: it is written that results for PBT are promising. It is noteworthy to mention in the discussion that the PBT studies presenting OS data are either very small prospective single institution trials with 10 patients or retrospective trials. This group also mixes stereotactic and normo-fractionated RT studies.
Response 4:
We agree with the reviewer. Therefore, we mentioned in page 1, line 42-44, that due to the limited availability of proton therapy and its associated costs, most of the available proton studies for LAPC are non-randomized studies, recruited a small number of patients, or were designed as retrospective studies.
In addition, in page 2, line 71-77, it was mentioned as a limitation of the systematic review, that the wide variation in total radiation doses and dose per fraction, particularly in SBRT and PBT, along with incomplete reporting of data, such as OS and toxicity, contribute to challenges in comparison of data and due to limited number of studies reporting on PBT, as was the number of patients included. Thus, the role of PBT in the treatment of LAPC is still to be established.
Since these facts were already reported in the original manuscript, we did not modify the manuscript.
Point 6:
Page 2, line 57 - 63: Although it was not the scope of this trial to assess different chemotherapy regimens, it is important to emphasize more the importance of the chemo-regimen FOLFIRINOX, how radiotherapy can be integrated into it, which role surgery might play and how future studies might look like.
Response 5:
Indeed, these aspects were thus far only illustrated in a limited manner. Therefore, an additional paragraph providing further information about induction chemotherapy followed by radio-chemotherapy was inserted in the Discussion (page, 2 lines: 63-70). Since the integration of radiotherapy into alternative regimens with chemotherapy and surgery was not the focus of the study, we did refrain from speculating about the future three-modality study designs.

Reviewer 2 Report
This manuscript provide reviews on Normo Or Hypo-Fractionated Photon or Proton Radiotherapy 2 in the Management of Locally Advanced Unresectable 3 Pancreatic Cancer. This is a very nice review that provide both information on photon and proton treatment of pancreatic cancer. This article is suitable to be published.
Author Response
We thank Reviewer 2 for the precious time reading our manuscript. Since Reviewer 2 had no concerns, we did not modify the manuscript.
Reviewer 3 Report
This paper deals with the comparison between the data from the different radiotherapy modalities currently used against the pancreatic cancer. While the aim of this study is very interesting and of great interest for radiation oncologists, the authors have notably used the median overall survival which may be discutable since the causes of final death are not necessarily linked to the RT modality itself and alone. Besides, they concluded that the data are not comparable which is not surprising with the relatively low number of studies and the diversity of responses of patients. I suggest that the authors investigate further their data by considering the dose to the tumor per each session but overall whether the healthy tissues are concerned or not and how (to relate them to the toxicity grade and the nature of the adverse reactions early/late). Again the approach described in figure 2 is good but similar figure should be shown for the dose per fraction, the interval of time between dose and much other isodoses parameters including the dose received by healthy tissues. The tumor tagreting is good in both SBRT and protontherapy but the bystander effect on healthy tissues or the doses applied to them may be very different. Once they have established the approach of figure 2 for all these parameters, then they could compare modalities by crossing OS and with these parameters.
Author Response
Point 1
This paper deals with the comparison between the data from the different radiotherapy modalities currently used against pancreatic cancer. While the aim of this study is very interesting and of great interest for radiation oncologists, the authors have notably used the median overall survival which may be discutable since the causes of final death are not necessarily linked to the RT modality itself and alone. Besides, they concluded that the data are not comparable which is not surprising with the relatively low number of studies and the diversity of responses of patients.
I suggest that the authors investigate further their data by considering the dose to the tumor per each session but overall whether the healthy tissues are concerned or not and how (to relate them to the toxicity grade and the nature of the adverse reactions early/late). Again, the approach described in figure 2 is good but similar figure should be shown for the dose per fraction, the interval of time between dose and much other isodoses parameters including the dose received by healthy tissues. The tumor targeting is good in both SBRT and proton therapy but the bystander effect on healthy tissues or the doses applied to them may be very different. Once they have established the approach of figure 2 for all these parameters, then they could compare modalities by crossing OS and with these parameters.
Response 1:
Indeed, correlating the incidence of toxicity (acute/late) with each radiation type is directly related to the fractionation and the type of radiation therapy either photon or proton. However, the studies collected in this systematic review have different evaluation systems for toxicity, some other studies do not report the evaluation system they used. In addition, most of the studies report the side-effects individually for each side-effect and do not report the total grade G3 toxicity, so it was not possible to make a separate evaluation as we did in Fig 2. Furthermore, the late toxicity was not reported in most studies, so cannot be evaluated for each group as well. Finally, since this was a systematic review of published data, we did not have access to individual radiation treatment plans and their dose distributions. Therefore, we could not modify the manuscript as suggested by Reviewer 3.

Round 2
Reviewer 3 Report
I stay very concerned about the authors'reply to my comments (even if it is fair) since it reveals again basic limitations of the intercomparison methodology and therefore to the conclusions itself. The authors should integrate their reply to a discussion chapter to show the actual limitations of their study, at least and conclude more rigorously , maybe with with again more cautiousness.
Author Response
Response to Reviewer 3 Comments round 2
Point 1
I stay very concerned about the authors ‘reply to my comments (even if it is fair) since it reveals again basic limitations of the intercomparison methodology and therefore to the conclusions itself. The authors should integrate their reply to a discussion chapter to show the actual limitations of their study, at least and conclude more rigorously, maybe with again more cautiousness.
Response point 1:
We regret we did not satisfactorily answer the Reviewer’s comment during the last revision. Since we fully agree with the reviewer, we added an additional paragraph in the discussion section (page 18, lines 72-81), where it now reads:
Moreover, the wide variation in total radiation doses and doses per fraction, particularly in those studies concentrating on SBRT and PBT, along with incomplete reporting of data, prohibits the direct comparison of the toxicity and outcome data of the three modalities. Even though there seems to be a direct correlation between the incidence of toxicity (acute or late) and the fractionation schedule and radiation modality, it was not possible to address this by statistical testing, since substantial amounts of data were missing and different toxicity scoring metrics were used. Finally, the number of studies reporting on PBT was limited, as was the number of patients included. Thus, the role of SBRT and PBT in the treatment of LAPC is still to be established.
Also, the Conclusion on Page 18 was slightly modified: PBT is a promising new treatment modality for LAPC patients, however, further clinical studies, ideally using different fractionation schedules and including photon regimen as comparator are needed.

Round 3
Reviewer 3 Report
The authors have reached my requirement to modulate their conclusion because of statistical constraints